# Brain Analysis with a Complex Network Approach in Stroke Patients Based on Electroencephalography: A Systematic Review and Meta-Analysis

**DOI:** 10.3390/healthcare11050666

**Published:** 2023-02-24

**Authors:** Borhan Asadi, Juan Nicolás Cuenca-Zaldivar, Noureddin Nakhostin Ansari, Jaime Ibáñez, Pablo Herrero, Sandra Calvo

**Affiliations:** 1Department of Physiatry and Nursing, Faculty of Health Sciences, IIS Aragon, University of Zaragoza, C/Domingo Miral s/n, 50009 Zaragoza, Spain; 2Grupo de Investigación en Fisioterapia y Dolor, Departamento de Enfermería y Fisioterapia, Facultad de Medicina y Ciencias de la Salud, Universidad de Alcalá, 28801 Alcalá de Henares, Spain; 3Physical Therapy Unit, Primary Health Care Center “El Abajón”, 28231 Las Rozas de Madrid, Spain; 4Research Group in Nursing and Health Care, Puerta de Hierro Health Research Institute—Segovia de Arana (IDIPHISA), 28222 Majadahonda, Spain; 5Research Center for War-Affected People, Tehran University of Medical Sciences, Tehran P.O. Box 14155-6559, Iran; 6Department of Physiotherapy, School of Rehabilitation, Tehran University of Medical Sciences, Tehran P.O. Box 14155-6559, Iran; 7BSICoS Group, IIS Aragón, Universidad de Zaragoza, 50018 Zaragoza, Spain; 8Department of Bioengineering, Imperial College, London SW7 2AZ, UK

**Keywords:** brain function network, electroencephalography, stroke

## Abstract

Background and purpose: Brain function can be networked, and these networks typically present drastic changes after having suffered a stroke. The objective of this systematic review was to compare EEG-related outcomes in adults with stroke and healthy individuals with a complex network approach. Methods: The literature search was performed in the electronic databases PubMed, Cochrane and ScienceDirect from their inception until October 2021. Results: Ten studies were selected, nine of which were cohort studies. Five of them were of good quality, whereas four were of fair quality. Six studies showed a low risk of bias, whereas the other three studies presented a moderate risk of bias. In the network analysis, different parameters such as the path length, cluster coefficient, small-world index, cohesion and functional connection were used. The effect size was small and not significant in favor of the group of healthy subjects (Hedges’g = 0.189 [−0.714, 1.093], Z = 0.582, *p* = 0.592). Conclusions: The systematic review found that there are structural differences between the brain network of post-stroke patients and healthy individuals as well as similarities. However, there was no specific distribution network to allows us to differentiate them and, therefore, more specialized and integrated studies are needed.

## 1. Introduction

Stroke represents one of the most common causes of disability with regards to its impact on functional limitations [1]. In addition, because of the aging population, the absolute number of strokes is expected to increase in the coming years [2]. Apart from the positive and negative clinical features that may appear after a stroke, this type of neural lesion is typically associated with alterations in the oscillatory brain activity that can be measured from the lesioned areas in the brain. Typically, lesioned brain regions present a slowing of rhythmic activity as compared to the contralesional side. This can be observed by computing the ratio between the power at low frequency (e.g., delta) and high frequency (alpha/beta/gamma) of spectral components in the EEG [3] Brain connectivity also undergoes changes, and this has been researched during the last years [4,5,6,7,8,9,10,11,12,13]. At present, we mainly rely on structural images of the brain affected areas to take clinical decisions and make predictions about evolution. Brain networks are a relevant feature of brain function, and these networks typically present drastic changes after a certain region gets damaged.

Taking all this into account, network analysis can help to improve the clinical characterization of patients with a stroke, since brain networks determine which areas of the brain are physically or functionally connected to support cognitive and behavioral functions during the brain’s rest/default state [14]. A “small-world index” (SW) network reproduces the competence of the brain networks to locally and globally process the flow of information [15]. In this regard, through the analysis of EEG graphs, it has been shown that ischemic stroke leads to a rearrangement of the information flow between the two hemispheres in the brain, which is frequency-dependent [13]. Regarding the interest in this field, the number of articles with the ‘Brain Network’ keyword has grown very fast since 1985, and this interest has increased exponentially in the last years, from 8387 research manuscripts published in 2016 to 14,256 in 2021 in Pubmed.

Studies on network topology highlight functional reorganization after stroke [16,17], suggesting that changes in the spontaneous functional architecture of the brain connectivity affecting function could be produced by ischemic lesions [18]. The complexity of functional brain connectivity can be studied using graph theory, a mathematical approach used to analyze networks that may be used to analyze the brain’s complex networks through simplified schemes of nodes and edges [19]. In this field, a network is a mathematical representation of a complex system in the real world and is defined by a set of nodes (vertices) and links (edges) between pairs of nodes. Different parameters define the properties of a network, each of which defines the network in some way. These parameters are used for network analysis and comparison between networks created from healthy humans and stroke patients. Brain network theory considers brain segregation as the network tendency to be organized in clusters and analyzes them using the local and global clustering coefficient (CC). The local CC calculates the local cohesion of a node with its neighbors [20]. Additionally, it provides an overall measure of the cohesion of the nodes in the whole network.

The average shortest path length (PL) parameter is also used to analyze brain integration. This parameter represents the network ability to exchange information between distant regions [15,16,17,18,19]. The measure of network small-world index (SW) is defined as the ratio between CC and PL [21,22]. The SW coefficient is used to describe the balance between the local connectedness and the global integration of a network. When SW is larger than 1, a network is said to have small-world index properties. SW organization mixes short PL and high CC.

The rapid development of imaging technology, such as computed tomography (CT), positron emission tomography, magnetic resonance imaging (MRI) and EEG, allows researchers to access latent knowledge about brain changes during stroke onset and the recovery process [23]. EEG provides a continuous, real-time and non-invasive measurement of brain function and provides new insights into brain pathophysiology after stroke [24,25,26]. This technology has the advantage of being widely available, having a low cost and providing a good compromise in terms of spatio-temporal resolution for superficial structures in the brain. EEG-based measures can provide neurophysiological biomarkers in the early pre-treatment phase that may be used to make short- and long-term predictions regarding the evolution of a patient, and that may even provide relevant information to determine optimal treatment strategies [3]. Moreover, EEG technologies have advanced significantly in recent years, making it possible to record signals unobtrusively without needing to move patients from their beds; it is also possible to test neurofeedback interventions based on the EEG signal and to modulate certain brain rhythms to study its potential therapeutic effect [3,27].

In EEG studies, functional connectivity between two signals can be evaluated by using a measure of general synchronization [28], such as EEG coherence [29]. The increase of EEG coherence can be interpreted as a functional coupling between two anatomically separated regions [30]. In addition, EEG coherence can also be used as a measure of binding by synchrony and is a proposed solution to the connecting problem [31], the way that the brain integrates signals separated in space and time [32].

Stroke leads to a large variability in clinical presentation and outcomes [33]. There is substantial heterogeneity in the procedures and tools used for outcome assessment after stroke, most of which are poorly validated [34]. Consequently, the heterogeneity in the use of outcome tools comprises the data quality [33]. No previous works looked into the brain network changes as an outcome post-stroke. To the authors’ knowledge, there are not any previous reviews of studies that have carried out brain network analysis of EEG in stroke patients. Because of this, in this systematic review, our objective is to review the research carried out on different stroke areas (Broca, cerebellum, middle cerebral artery, thalamus) and clinical phases (i.e., acute, sub-acute), by networking analysis of EEG and to perform a meta-analysis of the EEG-related outcomes in stroke patients.

## 2. Materials and Methods

### 2.1. Study Design

This systematic review was performed following the Preferred Reporting Items for Systematic Reviews and Meta-Analyses (PRISMA) statement [35] and was registered with the International Prospective Register of Systematic Reviews (PROSPERO), registration number (reference number: 285761).

### 2.2. Data Sources

The electronic databases PubMed, Cochrane and ScienceDirect were searched from their inception until October 2021. Search strategies were performed following the instructions described by Greenhalgh [36] (see Appendix A). The research question followed the PICO format: in adults who suffered a stroke episode (P = population), taking EEG from participants (I = intervention), comparing patients with stroke with healthy individuals (C = comparison) and analyzing outcomes related to brain network (O = outcome). Due to the limited number of cohort studies, cross-sectional studies and case reports were also considered.

### 2.3. Study Eligibility Criteria

Experimental studies were eligible for inclusion if they met the following criteria: (1) included patients with stroke, (2) used EEG tools for assessment, (3) used network analysis and (4) were published in English language. Rehabilitation studies, effect of physical therapy and/or effect of any other procedure on stroke were excluded. Two independent reviewers (BA and NC) performed the systematic search and screened the title and abstract of the articles, with duplicates being removed. The full text of eligible papers was carefully read to decide whether the eligibility criteria were met. The two reviewers compared their findings, and a third reviewer (SC) helped to solve any possible discrepancies if needed. After the selected articles were considered suitable for inclusion, one reviewer (BA) extracted the following information: (1) study author(s), (2) year of publication, (3) characteristics of participants (e.g., sample size, study groups, mean age, sex distribution, and types of strokes), (4) type of study, (5) type of method used to measure connectivity between nodes, (6) outcome measures, (7) assessment and intervention protocol and (8) main results. Then, a second reviewer (NC) verified the findings, and possible disagreements were discussed.

The Newcastle–Ottawa Scale was used to assess the risk of bias and the quality of nonrandomized studies in meta-analyses [37]. Three factors were considered to score the quality of included studies: (1) selection, including representativeness of the exposed cohort, selection of the non-exposed cohort, ascertainment of exposure and demonstration that at the start of the study the outcome of interest was not present; (2) comparability, assessed on the basis of study design and analysis, and whether any confounding variables were adjusted for; and (3) outcome, based on the follow-up period and cohort retention and ascertained by independent blind assessment, record linkage or self-report. The quality of the studies (good, fair and poor) was rated by awarding stars in each domain following the guidelines of the Newcastle–Ottawa Scale. A “good” quality score required 3 or 4 stars in selection, 1 or 2 stars in comparability and 2 or 3 stars in outcomes. A “fair” quality score required 2 stars in selection, 1 or 2 stars in comparability and 2 or 3 stars in outcomes. A “poor” quality score reflected 0 or 1 star(s) in selection, 0 stars in comparability or 0 or 1 star(s) in outcomes.

### 2.4. Data Analysis

For the statistical analysis, the program R Ver. 4.1.2 was used (R Foundation for Statistical Computing, Institute for Statistics and Mathematics, Welthandelsplatz 1, 1020 Vienna, Austria). In the articles showing the results with median and interquartile range, these were transformed into mean and standard deviation using the appropriate formulas [38,39]. The combination of groups and the average of the means and standard deviations were performed using the appropriate formulas [40]. Data were obtained by request from the authors and, when it was not possible, by extracting data from the graphs available in the articles (since these did not present them in table format) using the webplotdigitizer software [41].

A meta-analysis was carried out taking the small-world index (SW) as measure of brain networks. A random effects model was applied given the heterogeneity between the studies and analyzing the level of significance between the groups of patients and healthy participants through the standardized difference of means (SMD), based on the mean, standard deviation and sample size in each group. Heterogeneity was analyzed by estimating the between-studies variance τ^2^ (calculated with the DerSimonian–Laird estimator with Hartung–Knapp correction), with the Cochran Q test as well as with the I^2^ estimator being defined with the latter as: 0–30%, unimportant heterogeneity; 30–50%, moderate heterogeneity; 50–75%, large heterogeneity; and 75–100%, significant heterogeneity. The effect size was calculated with Hedges’g, defined as small effect below 0.2, moderate effect between 0.2 and 0.8 and big effect above 0.8.

Heterogeneity was assessed by detecting the following: (1) outlier studies, defined as studies with extreme effect sizes whose confidence interval does not overlap with the confidence interval of the pooled effect and differs significantly from the overall effect [42]; and (2) influencing studies, defined as those which have a large impact on the pooled effect or heterogeneity, regardless how high or low the effect is [42]. To analyze heterogeneity, the following were used: (1) influence graphs, which indicate the fit of the studies to the model; (2) Baujat plots, which detect studies that overly contribute to the heterogeneity in a meta-analysis ([43]); and (3) leave-one-out meta-analysis, which are forest plots with recalculated pooled effects, with one study omitted each time. In addition, a Graphic Display of Heterogeneity (GOSH) was used, which plots the pooled effect size on the *x*-axis and the between-study heterogeneity on the *y*-axis, which allows for looking for specific patterns or clusters with different effect sizes and amounts of heterogeneity [44].

Subgroup meta-analysis were performed to explore the detected heterogeneity depending on the mode to calculate SW brain networks: direct SW (SW calculated directly), ratio of the raw weighted clustering coefficient (Cw), weighted characteristic path length (Lw) or the ratio of raw global efficiency (Eg) to local efficiency (El). Publication bias was analyzed using the Begg and Eggers tests and standard and trim and fill funnel plots [45]. Finally, the overall and subgroup meta-analysis powers were calculated.

## 3. Results

The PRISMA flow diagram (Figure 1) illustrates the screening process. The electronic search identified 64 records, and 2 other results were added through alternative sources. After removing 2 duplicated articles, 47 manuscripts were excluded for not meeting the inclusion criteria. From the 17 results selected, 7 of them were removed for not meeting the eligibility criteria. Finally, a total of 10 studies were kept, 9 of which were cohort studies.

Table 1 summarizes the studies included in this systematic review investigating the analysis of brain networks in stroke patients [4,5,6,7,8,9,10,11,12,13]. These studies include research on stroke that has had effects such as hemianopsia [8] and Broca [5] in patients. In all the included studies, the presence of a lesion in the brain was observed, and in fact, the lesion caused a change in the network, which is discussed below. All included studies are case–cohort studies [4,5,6,7,8,9,10,12,13] except one, which was a cross-sectional study [11] that helps to understand the network change at the time of stroke and after. The networking methods in the studies were different, as well as the analysis performed, although they all had some similarities. In the network analysis, different parameters were used, which include the PL, CC, SW, cohesion and functional connection (FC). Each of the articles examined some parameters and were not consistent in this regard.

**Table 1 healthcare-11-00666-t001:** Data of the studies investigating the analysis of brain networks in stroke patients.

Study	Population	Control	Intervention	Outcome	Results
Liu, Shuang et al., 2016 [6]	30 acute thalamic ischemic stroke patients	30 healthy subjects	EEG in resting condition with eyes closed was recorded.	The functional connectivity was estimated with partial directed coherence (PDC) [46].	Compared to the control group, the stroke group showed a trend of weaker cortical connectivity and a symmetrical pattern of functional connectivity; that is, there was less information transfer between electrodes on the brain.
Vecchio, Fabrizio et al., 2019 [4]	30 patients with middle cerebral artery stroke and 11 with cerebellar stroke.	30 healthy subjects	EEG was measured in resting state (at least 5 min) with eyes closed with 19 electrodes in the International 10–20 system position and sampling rate fixed at 256 Hz.	Functional connectivity of EEG data was carried out with eLORETA. The eLORETA algorithm is a linear inverse solution for detection of the EEG signals’ source [47].	Beta2 and gamma small-world index were increased in the right hemisphere of patients with cerebellar stroke, respectively, compared to healthy subjects, while the alpha 2 small-world index was increased only in patients with middle cerebral stroke. Cerebellar stroke differed from MCA in that it did not cause reorganization of the alpha 2 network, whereas it caused reorganization of the high-frequency network in the beta 2 and gamma bands with small-world index enhancement.
Rutar Gorišek, Veronika et al., 2016 [5]	10 Broca’s patients	10 healthy subjects	The testing and EEG recordings were performed from 10 to 90 days (mean 54.4 ± SD 30.7) after the ischemic stroke.	Coherences were calculated by using the mscohere function in Matlab.	It was shown that the precise balance between task-related theta synchronization and desynchronization found in healthy subjects was severely disrupted in Broca’s patients, and functional networks in the theta frequency band were significantly altered in the patient group.Gamma desynchronization was widespread in healthy controls, but in Broca patients, task-related desynchronization was less in the right hemisphere, and functional networks in the gamma frequency band were significantly altered in the patient group.
Vecchio, Fabrizio et al., 2019 [7]	139 consecutive patients were enrolled in the acute phase of stroke	110 healthy subjects	The EEG recording was performed at rest, with closed eyes.	EEG functional connectivity analysis has been performed using the eLORETA.	When comparing the patients with the control group, there were significant differences, with higher levels of SW in the healthy subjects.A strong negative correlation was found between the NIHSS at follow-up and the small-world index gamma index in the acute phase, giving the SW gamma index a predictive weight for recovery.
Wang, Lei et al., 2012 [8]	7 stroke patients with hemianopia	7 healthy control subjects	EEG data were recorded with 30 scalp electrodes with the patient kept awake with eyes closed throughout the EEG recording for 2 min.	Phase synchronization index (PSI) [48] has been used.	For each case of the brain network with a different number of edges, the weighted clustering coefficient of the network of hemianopia stroke patients seems to be generally higher than that of the normal control group.Hemianopia stroke patients generally had a lower weighted characteristic path length than the control group.
Dubovik, Sviatlana et al., 2013 [9]	20 stroke patients	19 healthy participants	EEG was recorded with a 128-channel EEG system in an awake, resting condition with eyes closed.	The electromagnetic neural activity at each gray matter voxel was reconstructed with an adaptive spatial filter (beamformer)	Increased functional connectivity (FC) was observed in non-lesioned areas. These changes were mostly related to the alpha frequency band, and FC in the dysfunctional brain regions was consistently reduced in the alpha frequency band.
De Vico Fallani, Fabrizio et al., 2009 [10]	1 stroke patient	8 healthy subjects	EEG signals were recorded with a sampling frequency of 2048 Hz from 128 scalp electrodes.	Brain functional connectivity is achieved through the computation of task-related coherence.	The differences mainly involved the highest spectral contents (beta and gamma bands). In these bands, the global and local performances of the patient were statistically lower than the control subjects in the PRE (during the planning period) and EXE (movement execution) intervals.Network topology changes were particularly prominent in the beta band, which is already involved in motor tasks [45], as well as in the gamma band.
Vecchio, Fabrizio et al., 2017 [11]	A 72-year-old patient with stroke	Before and during a stroke attack	EEG Holter was recorded for evaluating signs of stroke-related epilepsy.	Magnitude squared coherence used (mscohere)	SW decreases in stroke and increases after stroke.SW decrease in the delta band and SW increase in the alpha bands.Coherence decreases during stroke and increases after stroke.
Fanciullacci, Chiara et al., 2021 [12]	33 unilateral post stroke patients in the sub-acute phase: cortico-subcortical (*n* = 18) and subcortical (*n* = 15)	10 healthy subjects	EEG was recorded for 10 min with a 10/20 EEG system in an awake, resting condition with eyes closed.	to explore interconnectivity between the ROIs, and intracortical lagged linear coherence was computed	In both groups of patients, compared to healthy subjects, there was an increase in the small-world index of the resting-state network in the θ band.β-band network measures differed significantly between stroke patients, with greater resolution and small-world index in patients with cortical involvement.
Caliandro, Pietro et al., 2017 [13]	30 patients with ischemic lesion	30 healthy subjects	The EEG recording was performed at rest, with eyes closed and no task condition for at least 5 min from 19 electrodes.	Connectivity analysis using eLORETA in both hemispheres.	Resting-state network changes were mainly detected in low- and medium-frequency EEG bands, i.e., delta, theta and alpha 2 rhythms, while no network reorganization was found in alpha 1, beta and gamma bands.

The evaluation of the quality and risk of bias of cohort studies is shown in Table 2; five studies were of good quality [4,5,7,8,13] and four were of fair quality [6,9,10,11].

The risk of bias analysis is shown in Figure 2. Six studies showed a low overall risk of bias [5,6,7,9,12,13]. However, three studies presented an overall moderate risk of bias due to some concerns about outcome measures and reported outcomes [4,8,10]. The overall risk of bias is low to moderate due to missing data and the selection of reported studies.

Only one of the studies [11] assessed was a cross-sectional study (see Table 3). The study [11] included in this review does not have a sufficiently defined protocol, and the analysis was not repeated by more than one researcher to ensure reliability. However, it has many positive strengths that are indicated in Table 3 with the answer “yes”.

### Meta-Analysis

The presence of relatively high values for τ^2^ (0.25), the significant Cochrane Q test (*p* = 0.04) and the value of I^2^ of the 60% indicate large heterogeneity. The effect size is small and not significant in favor of the group of healthy subjects (Hedges’g = 0.189 [−0.714, 1.093], Z = 0.582, *p* = 0.592) (Figure 3).

No study analyzed is an outlier that exceeds the 95% CI (Appendix A, Figure A1). The influence analysis shows how the study published by Fanciullacci (red dot) exerts a great influence on the result of the meta-analysis (Appendix A, Figure A2). The Baujat graph shows again that the Fanciullacci study (upper right square dot) also exerts an excessive influence on the heterogeneity detected (Appendix A, Figure A3). The sensitivity analysis to the withdrawal of a study indicates, both by effect size and by I2 values, that the study published by Fanciullacci is the one that most influences the result of the meta-analysis (Appendix A, Figure A4). The GOSH graph shows the presence of two clusters of studies with a heterogeneity around 60–80% and 0–20% respectively, with an effect that oscillates around 0 and 0.5 in both (Appendix A, Figure A5).

When the meta-analysis of subgroups was carried out, we found that there is a decrease in heterogeneity in the studies carried out with Raw Cw, Lw (I^2^ changes from 60% to 0%), whereas there is an increase in the studies carried out with Direct SW (I^2^ changes from 60% to 79%). Therefore, it is possible that studies with Direct SW are responsible for part of the heterogeneity detected. We verified how the effect size in the studies carried out with Raw Cw-Lw is small and not significant in favor of the group of healthy subjects (Hedges’s g = 0.008, [−2.229, 2.245], Z = 0.047, *p* = 0.97). In the group of studies carried out with Direct SW, the effect size is moderate and not significant and displaced in favor of the group of healthy subjects (Hedges’s g = 0.535, [−6.76, 7.831], Z = 0.932, *p* = 0.522). Finally, in the study carried out with Raw Eg-El, the effect size is large and not significant and displaced in favor of the patient group (Hedges’s g = −1.431, [−3.416,0.555], Z = −1.412, *p* = 0.158). Therefore, it can be concluded that none of the study groups produced significant effects. Despite the disparity of the effect in each group (0.008 vs. 0.535 vs. −1.431), there are no significant differences in the effect between groups depending on the calculation mode of the SW (X2(2) = 2.86, *p* = 0.239) (Figure 4).

The tests of Begg (Kendall’s τ = 0, *p* = 1) and Eggers (t(3) = −1.187, *p* = 0.321) were not significant and indicate the absence of publication bias. The funnel plots (both the standard and the one obtained with trim and fill method) show a symmetrical distribution except for the study carried out by Fallani, which corroborates the absence of publication bias (Appendix A, Figure A6).

## 4. Discussion

Studying cortical connectivity changes in stroke can provide novel ways to characterize clinical rehabilitation and suitable therapies for patients with brain lesions leading to cognitive or motor disability. Here, we revise the existing literature supporting the use of EEG-based metrics to analyze brain networks and how they are altered in stroke patients. Despite the identified differences between the studies considered here in terms of recording conditions (e.g., devices or number of channels used) and metrics used to assess cortico-cortical connectivity, it was possible to find a common conclusion in them: the brain network of stroke patients was shown to be different from that of healthy people. These differences were only found in specific frequency bands. For example, there were neither significant differences between post-stroke patients with hemianopia and the healthy controls in the global weighted CC and PL [8] parameters nor in the network rearrangement in alpha 1, beta and gamma bands [13].

In all the studies analyzed, the internal structure of the brain network changed compared to healthy humans. The greatest impact was observed in the affected areas, which showed weaker connections. If we consider delta, theta and alpha bands as low-frequency bands, and assume beta and gamma as high-frequency bands, the functional and topological network in high-frequency bands in patients after an acute stroke show changes [4,5,7,10,12,13]. However, no conclusion can be drawn with certainty about the SW index: in some studies, the patients had a higher SW in different bands [4,12], whereas in others [11], the SW index increased after stroke. However, another study showed that the SW parameter was higher in the control group [7]. It could be expected that healthy patients have a higher SW so that transfer data in the brain is done faster from one point to another, following the shortest path length, but this is still unclear, and data obtained in different populations are still controversial.

As in the case of the small-world index, no statistically significant differences were found in indices, such as the shortest path length and clustering coefficient, although recently a case study showed significant changes in these parameters when the analysis was performed by dynamic complex network, which shows higher variations in the global clustering coefficient and small-world index and lower variations in the average shortest path length in the low-frequency bands, such as delta, theta and alpha bands, in the chronic stroke patients [49].

Although evidence suggests that connections are less in the brain of post-stroke patients, a model of brain network cannot be clearly considered, or there is not a standard by which patients with stroke can be recognized. However, the immediate EEG changes observed after stroke are a direct consequence of cerebral blood flow reduction that later results in neuronal impairment or neuronal death. This neuronal impairment in turn leads to a disorganization of the electrical activity that is reflected by the global EEG changes [50]. Subsequent neuroplastic changes after stroke have been largely reported [51]. Post-stroke neuroplasticity can lead to reorganization in neural circuits that allow for regaining the lost functionality [52,53]. However, according to our knowledge, there have not been any publications discussing how neuroplasticity after a stroke leads to changes in brain network parameters measured with EEG.

Oscillating neural activity in the gamma frequency band is involved in several cognitive functions, including visual object processing [54,55], attention [56,57] and memory [58,59]. Additionally, several studies have demonstrated that gamma band activity is strongly associated with behavioral performance in several memory tasks [60,61] and that there is a higher gamma band activity in participants exhibiting superior recognition memory performance [62]. Other findings [63,64] suggest that gamma oscillations not only reflect brain activity related to memory processes, but also vary with the memory ability of individuals. Two of the studies included in the review [5,7] described variation in the gamma band, which could be supported by evidence that suggests that gamma oscillations mediate information transfer between cortical and hippocampal structure for memory abilities [58,59].

With some exceptions, a consensus is reached on how an increase in the slow band frequencies, referred to as slow oscillation and delta oscillation, and is associated with not only the slow-wave sleep state, but also brain ischemia. Conversely, high band frequencies, such as the alpha, beta and gamma oscillations, are associated with wake states or cognitive task engagement, and their presence frequently reduces after stroke. As it was found in one study [10], changes were seen in gamma and beta bands. A possible explanation could be that coherence in higher bands may be more involved in active (either motor or cognitive) tasks [30,31,32].

In our review, differences in EEG signals were observed not only in parameters, but also in different bands. For example, in one study [13], changes were seen in low-frequency bands (delta, theta and alpha2), and no difference was seen in higher-frequency bands (alpha1, beta and gamma), which is consistent with the results of a recent case study [49]. In other studies, changes were seen in alpha [7,9] and theta and beta bands [12]. The changes in the different bands that are seen can be related to the type of stroke, the type of networking of the brain regions, the resting or active state, the examined area of the brain and the age of the patient [65].

Regarding the usefulness of EEG-based connectivity measures to assess brain function in stroke, we found some limitations related to the heterogeneity of the injured areas in the patients participating in the studies included in our meta-analysis. Future studies should include larger samples and grouping patients in terms of the exact locations of their lesions, which would allow us to further understand how the spatial location of a brain lesion affects the rearrangement of brain connections at the local and global scales.

## 5. Conclusions

This is the first systematic review carried out about EEG connectivity changes to diagnose or characterize stroke. The systematic review found that there are structural differences between the brain network of post-stroke patients and healthy individuals as well as similarities. However, there is no specific distribution network that allows us to differentiate them and, therefore, more specialized and integrated studies are needed.

## Figures and Tables

**Figure 1 healthcare-11-00666-f001:**
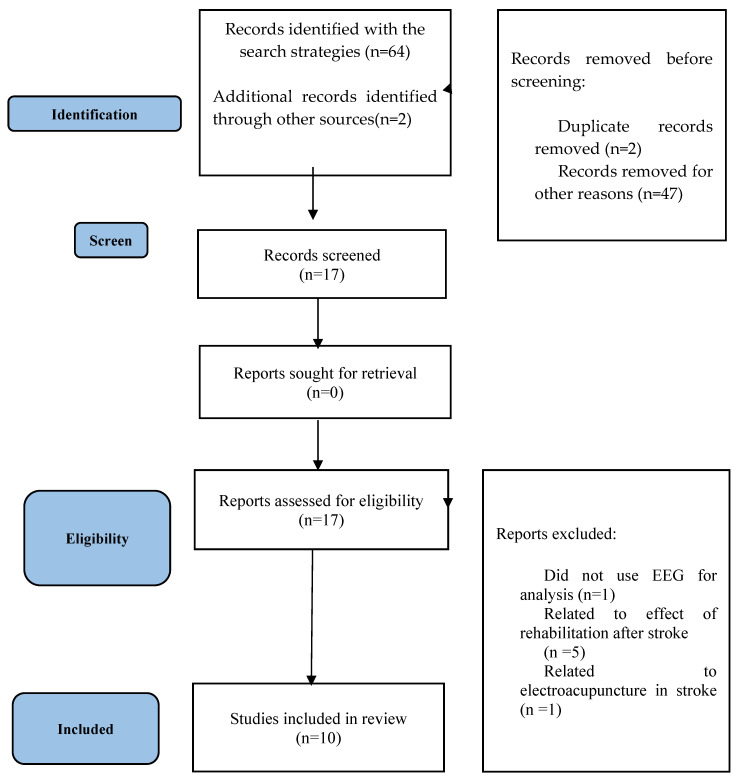
Preferred Reporting Items for Systematic reviews and Meta-Analyses (PRISMA) flowchart.

**Figure 2 healthcare-11-00666-f002:**
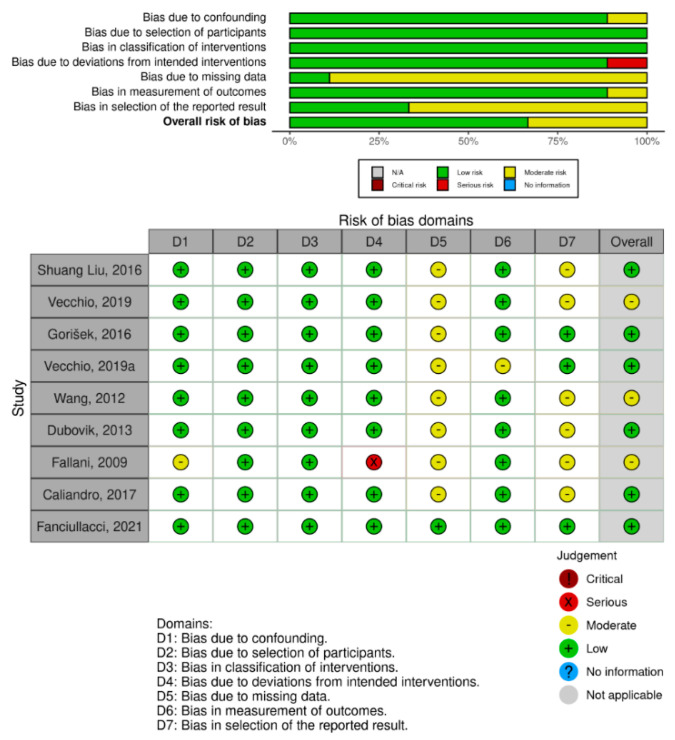
Risk of bias (RoB) traffic light plot. Assessment of the risk of bias via the traffic light plot of the RoB of each included clinical trial and the weighted plot for the assessment of the overall risk of bias via the Cochrane Robins-I tool (*n* = 9 studies). Yellow circle indicates some concerns on the risk of bias, and green circle represents low risk of bias [4,5,6,7,8,9,10,12,13].

**Figure 3 healthcare-11-00666-f003:**
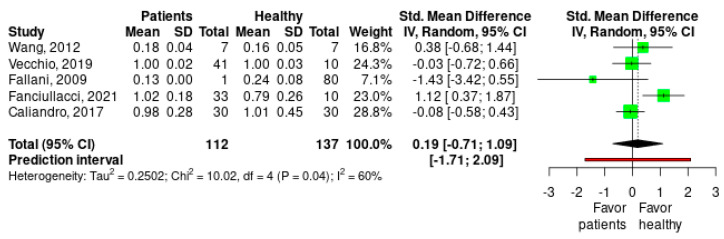
Meta-analysis forest plot for small-world index brain network measure on healthy subjects vs. stroke patients. SD: standard deviation; SMD: standardized mean difference; 95%CI: 95% confidence interval; Tau2: τ^2^ between studies variance estimation; Chi^2^: Cochran Q test; df: degrees of freedom; I^2^: proportion of the variance in observed effect is due to variance in true effects rather than sampling error [7,8,10,12,13].

**Figure 4 healthcare-11-00666-f004:**
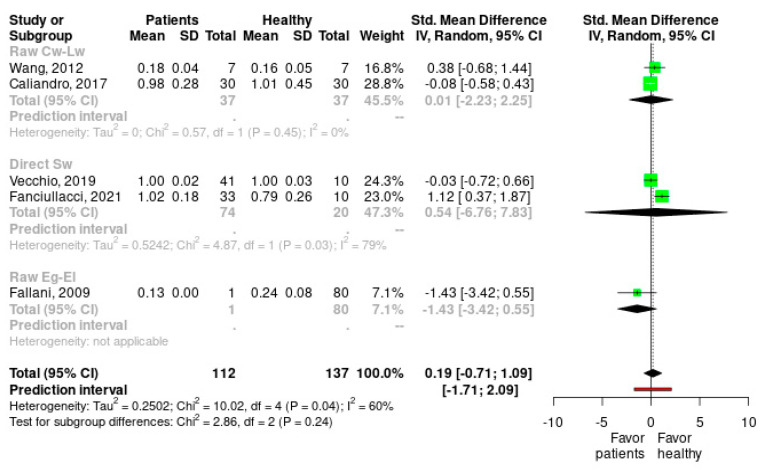
Subgroup forest plots for the different calculation methods of small-world index on healthy subjects vs. stroke patients. Cw: weighted clustering coefficient; Lw: weighted characteristic path length; Eg: global efficiency; El: local efficiency. SD: standard deviation; SMD: standardized mean difference; 95%CI: 95% confidence interval; Tau^2^: τ^2^ between studies variance estimation; Chi^2^: Cochran Q test; df: degrees of freedom; I^2^: proportion of the variance in observed effect is due to variance in true effects rather than sampling error [7,8,10,12,13].

**Table 2 healthcare-11-00666-t002:** Risk of bias assessment (Newcastle–Ottawa Quality Assessment Scale criteria).

Study	Selection				Comparability	Outcome			Quality Score
	Representativeness of Exposed Cohort	Selection of the Non-Exposed Cohort from Same Source as Exposed Cohort	Ascertainment of Exposure	Outcome of Interest Was Not Present at Start of Study	Comparability of Cohorts	Assessment of Outcome	Follow-up Long Enough for Outcome to Occur	Adequacy of Follow-Up	
Liu, Shuang et al., 2016 [6]	Participants were in two groups: ischemic thalamic stroke (*n* = 30) and the healthy group (*n* = 30). ★	Yes ★	Inclusion criteria of the patients consisted of focal ischemic lesion of the thalamus and hand numbness as symptoms.	Yes ★	Nothing matched	Comparison of parameters of brain network between ischemic thalamic stroke patients and healthy group.	Yes ★	All stroke patients from whom EEG was taken participated in the study. ★	Fair
Vecchio, Fabrizio et al., 2019 [4]	Patients were in two groups: cerebellar and middle cerebral artery strokes (*n* = 30) and healthy group (*n* = 30). ★	Yes ★	The patients were clinically assessed by the National Institutes of Health Stroke Scale (NIHSS) during the acute phase.	Yes ★	Age and gender matched ★	Comparison of parameters of brain network between stroke patients and healthy group.	Yes ★	All stroke patients from whom EEG was taken participated in the study. ★	Good
Rutar Gorišek, Veronika et al., 2016 [5]	Participants were in two groups: Broca’s patients (*n* = 10) and healthy group (*n* = 10). ★	Yes ★	Boston Diagnostic Aphasia Evaluation (BDAE)	Yes ★	Sex and education matched ★	Comparison of parameters of brain network between stroke patients and healthy group.	Yes ★	All patients from whom EEG was taken participated in the study. ★	Good
Vecchio, Fabrizio et al., 2019 [7]	Participants were in two groups: patients with stroke in the acute phase (*n* = 139) and healthy group (*n* = 110). ★	Yes ★	All patients were clinically evaluated with three scales for stroke: NIHSS, Barthel and ARAT.	Yes ★	Sex and age matched ★	Comparison of parameters of brain network between stroke patients and healthy group.	Yes ★	All patients from whom EEG was taken participated in the study. ★	Good
Wang, Lei et al., 2012 [8]	Participants were in two groups: stroke patients (*n* = 7) and healthy controls (*n* = 7). ★	Yes ★	All patients were diagnosed with hemianopia stroke according to visual threshold test and MRI/CT scanning.	Yes ★	Sex and age matched ★	Comparison of parameters of brain network between stroke patients and healthy group.	Yes ★	All patients from whom EEG was taken participated in the study. ★	Good
Dubovik, Sviatlana et al., 2013 [9]	Participants were in two groups: patients with ischemic stroke (*n* = 20) and healthy participants (*n* = 19). ★	Yes ★	Motor function was evaluated by means of the Jamar dynamometer, the Nine Hole Peg test, the Stroke Rehabilitation Assessment of Movement (STREAM) and the Fugl–Meyer score.	Yes ★	Age matched	Assessment resting-state functional connectivity with (EEG).	Yes ★	All patients from whom EEG was taken participated in the study. ★	Fair
de Vico Fallani, Fabrizio et al., 2009 [10]	Participants were in two groups: healthy subjects (*n* = 8) and one patient with stroke. ★	Yes ★	No information	Yes ★	Nothing matched	Analysis of cerebral electro-physiological activity during planning or execution of movement in in stroke patients.	Yes ★	All patients and healthy people from whom EEG was taken participated in the study. ★	Fair
Fanciullacci, Chiara et al., 2021 [12]	Participants were in two groups: stroke patients in the sub-acute phase (*n* = 33) and healthy subjects (*n* = 10). ★	Yes ★	Brain injury was assessed by means of a standard CT scan.	Yes ★	Age matched	Characterizing resting-state EEG activity and functional connectivity changes in a cohort of unilateral ischemic patients compared with the healthy group.	Yes ★	All patients from whom EEG was taken participated in the study. ★	Fair
Caliandro, Pietro et al., 2017 [13]	Participants were in 2 groups: patients with ischemic lesion (*n* = 30) and healthy subjects (*n* = 30). ★	Yes ★	Patients were clinically evaluated by the National Institutes of Health Stroke Scale.	Yes ★	Age and sex matched ★	Whether and how ischemic stroke in the acute stage may determine changes in the small-world index of cortical networks.	Yes ★	All patients from whom EEG was taken participated in the study. ★	Good

★ The quality of the studies (good, fair and poor) was rated by awarding stars in each domain following the guidelines of the Newcastle–Ottawa Scale. A “good” quality score required 3 or 4 stars in selection, 1 or 2 stars in comparability and 2 or 3 stars in outcomes. A “fair” quality score required 2 stars in selection, 1 or 2 stars in comparability and 2 or 3 stars in outcomes. A “poor” quality score reflected 0 or 1 star(s) in selection, 0 stars in comparability or 0 or 1 star(s) in outcomes.

**Table 3 healthcare-11-00666-t003:** Quality assessment for cross-sectional study.

Vecchio, Fabrizio et al., 2017 [11]	
Did the study address a clearly focused question/issue?	Yes
Is the study design appropriate for answering the research question?	Yes
Does the study have a well-defined protocol?	No
Are both the setting and the subjects representative with regard to the population to which the findings will correlate?	Yes
Is the researcher’s perspective clearly described and taken into account?	Yes
Are the methods for collecting data clearly described?	Yes
Are the methods for analyzing the data likely to be valid and reliable? Are quality control measures used?	Yes
Was the analysis repeated by more than one researcher to ensure reliability?	No
Are the results credible, and if so, are they relevant for practice? Are results easy to understand?	Yes
Were there clinically relevant outcomes?	Yes
Are the conclusions drawn justified by the results?	Yes
Are the findings of the study transferable to other settings?	No

## Data Availability

The data that support the findings of this study are available from the first author (B.A.), upon reasonable request.

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
