# Peer review of "Brain Analysis with a Complex Network Approach in Stroke Patients Based on Electroencephalography: A Systematic Review and Meta-Analysis"

_healthcare, 2023, doi:10.3390/healthcare11050666_

Round 1
Reviewer 1 Report
This is an interesting and potentially important review. The authors performed a systematic review and meta-analysis on the complex EEG network approach in stroke patients. The outcomes of the review highlight the existing structural differences in brain networks in post-stroke patients, as well as some similarities with healthy controls.
The aim of the review is relevant and the paper is overall well structured.
However, I have some concerns:
Introduction:
The authors in the introduction mainly focused on network analysis methods, but the stroke-related EEG spectral alterations in stroke acute and chronic phases are not sufficiently reported. This should be the starting point for further complex network analysis.
Lines 62-83
Too many details about graph theory. I invite authors to be more concise, possibly without formulas.
Lines 87 - 88
Meanwhile, EEG provides a continuous, real-time, and non-invasive measurement of brain function and provides new insights into brain pathophysiology after stroke [23-25]
Please consider the recent literature on this topic, taking into account the technological progress in the field that allowed improved EEG assessments and studies (such as: 10.1007/s10439-021-02735-w , 10.1161/01.STR.0000122622.73916.d2 , 10.1007/s11517-020-02280-z , 10.1016/j.clinph.2007.07.021 , 10.1016/j.clinph.2018.05.021)
Line 97
Stroke leads to a large variability of clinical outcomes [31], with recent reviews pointing out that there have been found more outcomes than trials [32]
The second part of the sentence is not clear.
Lines 100 - 120
Because of it, in this systematic review, our objective is to review the research carried out on different types of strokes by networking analysis of EEG and to perform a meta-analysis of the EEG-related outcomes in stroke patients.
Please be more clear about different types of strokes.
Methods
Did the authors take into account factors such as stroke severity and stroke location?
Results
The results contain many figures and tables, making the work harder to read. Consider moving figures and tables to the supplementary section, and I invite authors to summarize the section to point out the most important findings.
Discussion
Lines 320 -324
However, the immediate EEG changes observed after stroke are a direct consequence of cerebral blood flow reduction that later results in neuronal impairment or neuronal death. This neuronal impairment in turn leads to a disorganization of the electrical activity that is reflected by the global EEG changes 323 [48].
I invite the authors also to discuss the effect of neuroplasticity on the disorganization/reorganization of electrical activity.
Author Response
We attach the point by point reply. We think the manuscript has improved with your review and we would like to thank you for dedicating your time to review. We hope you find our manuscript suitable for publication

Reviewer 2 Report
In their paper, Asadi et al., conducted the systematic review of the complex brain networks obtained using electroencephalography in stroke patients. The graph theoretical metrics such as clustering coefficient, path length, small-world, and cohesion were used. The results showed a functional network differences between post-stroke patients and healthy controls. The results were very interesting but after thoroughly reading the manuscript I have some concerns. I would propose the minor revision of this manuscript and please see my specific comments below.
· The location of stroke plays a vital role in the effects of stroke. Does the authors consider the location of stroke in the analysis?
· The introduction section needs more focus on stroke and graph theoretical formulas should be moved to the methods section.
· All the figure legends need to be self-explanatory and must include the abbreviations used in that specific figure.
· Please use small world index in the entire manuscript instead of small world (For example, line 306).
· The reason for choosing the specific graph metrics is not clear.
Author Response

(The authors gave the same response as above.)

Round 2
Reviewer 1 Report
Thank you for addressing all the issues raised in the previous review. I have carefully read the revised manuscript, and I am pleased to confirm that all of my concerns have been satisfactorily addressed. The additional data and explanations provided have significantly improved the manuscript.
I am satisfied that the revised manuscript is now suitable for publication and will be of great interest to readers interested in brain network analysis in stroke patients. I appreciate the effort and attention to detail that went into this research, and I look forward to seeing more work from this team in the future.